# Sustainable Bio-Based UV-Cured Epoxy Vitrimer from Castor Oil

**DOI:** 10.3390/polym15041024

**Published:** 2023-02-18

**Authors:** Matteo Bergoglio, David Reisinger, Sandra Schlögl, Thomas Griesser, Marco Sangermano

**Affiliations:** 1Politecnico di Torino, Department of Applied Science and Technology, Corso Duca degli Abruzzi 24, 10129 Torino, Italy; 2Polymer Competence Center Leoben GmbH, Roseggerstrasse 12, 8700 Leoben, Austria; 3Institute of Chemistry of Polymeric Materials, Montanuniversitaet Leoben, Otto Glöckel-Straße 2, 8700 Leoben, Austria

**Keywords:** vitrimeric network, sustainable thermoset, biobased epoxy, dynamic characteristics, exchangeable bonds

## Abstract

Vitrimers brought new properties in thermosets by allowing their reshaping, self-healing, reprocessing, and network rearrangement without changing structural integrity. In this study, epoxidized castor oil (ECO) was successfully used for the straightforward synthesis of a bio-based solvent-free vitrimer. The synthesis was based on a UV-curing process, which proceeded at low temperatures in the absence of any solvents, and within a short time. Real time Fourier-transformed infrared spectroscopy and photo-DSC were exploited to monitor the cationic photocurable process. The UV-cured polymer networks were able to efficiently undergo thermo-activated bond exchange reactions due to the presence of dibutyl phosphate as a transesterification catalyst. Mechanical properties, thermal resistance, glass transition temperature, and stress relaxation were investigated as a function of the amount of transesterification catalyst. Mechanical properties were determined by both DMTA and tensile tests. Glass transition temperature (*T_g_*) was evaluated by DMTA. Thermal stability was assessed by thermogravimetric analysis, whilst vitrimeric properties were studied by stress relaxation experiments. Overall, the ECO-based vitrimer showed high thermal resistance (up to 200 °C) and good mechanical properties (elastic modulus of about 10 MPa) and can therefore be considered as a promising starting point for obtaining more sustainable vitrimers.

## 1. Introduction

Conventional thermosets are highly attractive in today’s market due to their excellent characteristics, including thermal stability, chemical resistance, and excellent mechanical properties [1,2,3]. Some of the most used thermoset resins correspond to unsaturated polyester, epoxy resin, phenolic resin, and isocyanates, which are derived from petroleum-based resources [4,5].

The interconnected structure, by crosslinking bonds, donates the numerous merits that brought the research to use thermoset materials in the fields of coatings, composites, adhesives, and fiber-reinforced composites [6]. However, a tangible consequence of their optimal properties gives a critical drawback, resulting in difficult recycling. Heating a thermoset would bring degradation of the network instead of reaching an optimal viscosity for reprocessing, which happens in thermoplastic materials. Conventional thermosets waste is handled through re-grinding to use it as filler without changing the structure or combusted to recover energy. However, these types of recovery are considered the last choice in the waste management approach by European Union’s Waste Framework Directive [7]. It is therefore essential to find new recycling methods for thermosets that include chemical recycling of the structure, avoiding significant environmental burdens, high recycling costs, and energy consumption.

One interesting strategy to obtain this fundamental recycling goal is the exploitation of labile covalent and reversible dynamic bonds present in Covalent Adaptive Networks (CANs), crosslinked polymer networks designed by Bowman and co-workers [8]. CANs can be used in thermoplastic and thermoset materials. Mainly, they are exploited in thermosets. CANs, following an associative bond exchange mechanism, have been coined vitrimers by Leibler and co-workers in 2011 to indicate the crosslinked polymer network that exhibits a decrease of viscosity upon heating [9]. Dynamic bonds lead to new properties in the thermosets that, besides their desired characteristics, can overcome their drawbacks, allowing reshaping, self-healing properties, reprocessing, and network rearrangement without loss of structural integrity [10,11]. Many dynamic associative exchange reactions have been used in vitrimers.

Vitrimers are therefore covalently linked polymer networks containing dynamic bonds which can be exploited to change the topology of the covalent bond in the network, via exchange reaction, always keeping constant the number of chemical bonds. Upon heating above a certain temperature, vitrimers behave like viscoelastic liquid without being dissolved in good solvents. The constant number of covalent bonds allows the material to remain insoluble, but shows a gradual Arrhenius-type viscosity change in temperature. The gradual decrease in the viscosity allows the crosslinked polymer network to be reshaped and recycled.

Carboxylate transesterification is particularly interesting as dynamic bonds because, with the aim of a catalyst, it is possible to generate rapid exchange reactions, especially in epoxy-based resins. Exchange reactions allow the rearrangement of the chemical structure and the possibility of deforming, reprocessing, and recycling the material. Carboxylate transesterification uses both hydroxyl and ester groups [12,13,14,15,16].

Another way to increase the sustainability of thermoset materials is to combine the sustainability of CANs with bio-based monomers for their synthesis. The use of bio-based polymers, monomers, and oligomers can represent a green alternative since they are based on renewable resources and possess low CO_2_ emissions. Many options of bio-derived molecules derived from agro-food industrial wastes are available [14,17,18,19], and they have been exploited for curing processes.

The conventional curing method to obtain vitrimers is the thermal curing process, which requires a long curing time and high energy consumption. For these reasons, although the final bio-based and reprocessable thermoset is environmentally friendly, the curing process is not optimal for obtaining a completely sustainable material. To overcome this issue, photochemistry can be helpful. The UV-curing process is low energy consuming since it uses UV irradiation, which is generated with less energy than heating and requires only minutes to complete. Photo-induced processes can be divided into two categories based on their mechanism: radical and cationic [20,21,22]. The cationic polymerization is promising for curing epoxy monomers, and frontal process can also be exploited to achieve crosslinking in thick samples, as demonstrated in many articles [23,24,25,26,27,28,29]. The process involves the propagation of a reactive carbocation species that is generated after the photo-cleavage of an iodonium salt (photo-initiator) that, in turn, generates a superacid able to start the cationic ring-opening epoxy polymerization, hence the reactive carbocation [22].

In our group we have deeply investigated the sustainability of UV-curable process by studying the reactivity of bio-based monomers towards both radical [30,31] and cationic [22,32,33,34,35,36,37] mechanism. We have also studied fossil-based photo-cured vitrimers, where we described a convenient approach for a locally controllable photoactivation of vitrimeric properties in a covalently crosslinked thiol-epoxy network, exploiting the UV-mediated release of a strong amidine base acting as an efficient transesterification catalyst [38].

There is plenty of literature on the use of bio-based monomers for vitrimers preparation [11,39,40,41]. Between the bio-based monomers used as a starting point for the production of vitrimers, castor oil is gaining interest thanks to the versatile reactive groups that contain and can be used in the transesterification reaction. In fact, castor oil structure contains ester groups, hydroxyl groups, and unsaturated carbon-carbon bonds, which allows for modifying it to tailor its properties since the non-conjugated double bonds are not reactive enough to induce a direct free radical polymerization [42,43]. For the reasons explained, castor oil has been modified in many kinds of research to incorporate epoxy, amine, and methacrylate reactive groups [43,44,45,46]. Epoxidized castor oil (ECO) is synthetized from castor oil using a formic acid/H_2_O_2_ system via epoxidation. The synthesis process is efficient, inexpensive, and has a high yield [47]. ECO can be used as a plasticizer, stabilizer, and to produce bio-based epoxy resin [46,48,49].

In this study, we used ECO as the only precursor material to produce a photo-cured, solvent-free, bio-based vitrimer. ECO was photo-cured using triaryl sulfonium hexafluorophosphate salt (THS) as a photo-initiator, while the transesterification reaction was catalyzed through dibutyl phosphate. The reactivity of the different formulations was investigated by means of ATR-FTIR and photo-DSC measurements. The UV-cured films were fully characterized and the vitrimeric properties were assessed by stress relaxation studies.

## 2. Materials and Methods

### 2.1. Materials

Epoxidized castor oil (ECO) was purchased from specific polymers, and dibutyl phosphate (DP) and triaryl sulfonium hexafluorophosphate salt (THS) were purchased from Sigma Aldrich, Milano, Italy.

### 2.2. Formulation and Photo-Curing Process

ECO bio-based resin was mixed with cationic photo-initiator and variable amounts of transesterification catalyst. The three components, castor oil, photo-initiator, and transesterification catalyst, were mixed through an ultrasonic bath until all the components were homogeneously dissolved. The formulations were stored in dark environment to avoid early curing due to light contact before pouring them into silicon molds, and subsequently, they were UV-cured using DYMAX ECE Flood lamp (Dymax Europe GmbH, Wiesbaden, Germany) at a light intensity of 130 mW/cm^2^ for 60 s.

### 2.3. Characterization Methods

#### 2.3.1. Attenuated Total Reflectance Fourier Transform Infrared Spectroscopy (ATR-FTIR)

The crosslinking reaction was followed through a Nicolet iS 50 Spectrometer. The analysis was made after spreading the liquid formulation on a SiC substrate. The thickness sample was 32 µm, which was reached using a stir bar. The spectral resolution of the spectra was 4 cm^−1^ and the data were elaborated by using OMNIC software from Thermo Fisher Scientific.

The conversion results were obtained by following the decreasing of the epoxide peak centered around 850 cm^−1^, and the peak at 2950 cm^−1^ was taken as reference, since the C-H stretch was considered unaffected by UV-light. In order to determine the conversion behavior, Equation (1) was used during the exposure time.
(1)$$Conversion\;\left(\%\right)=\frac{{\left(\frac{A_{group}}{A_{ref}}\right)}_{t=0}-{\left(\frac{A_{group}}{A_{ref}}\right)}_{t}}{{\left(\frac{A_{group}}{A_{ref}}\right)}_{t=0}}\times 100$$
where *A_group_* is the area of the selected group under investigation and *A_ref_* is the reference area of the peak situated at 2950 cm^−1^. The study was done during exposure time in order to elaborate the conversion curve.

#### 2.3.2. Photo Dynamic Scanning Calorimetry (Photo-DSC)

The crosslinking reaction was followed through a photo-DSC measurement. The instrument was composed by a Mettler TOLEDO DSC-1 equipped with Gas Controller GC100 and a mercury lamp, Hamamatsu LIGHTINGCURE LC8 (Hamamatsu Photonics), used with an optic fiber in order to focalize the radiation.

The wavelength of UV-light was settled at 365 nm, and the intensity was 10% of the maximum intensity, resulting in 100 mW/cm^2^. 5 to 15 mg of samples were placed in an open aluminum pan, and an empty aluminum pan was taken as a reference. All the tests were measured under 40 mL/min of N_2_ flow at room temperature (25 °C).

The evaluation method involved the settling of the sample for two minutes and then two expositions to the UV-light of ten minutes each. The second exposition needed to be certain of a total conversion of the group and to create the baseline, in fact, the second curve was subtracted from the first one to obtain only the curve related to the reticulation. All the data were elaborated by Mettler Toledo STARe software V9.2.

#### 2.3.3. Dynamic Mechanical Thermal Analysis (DMTA)

A thermomechanical analysis of the thermosets was performed using a Triton Technology device. The initial temperature was set at −30 °C, temperature reached through using liquid nitrogen, and the measure was stopped at 100 °C, with a heating rate of 5 °C/min. The device applied uniaxial tensile stress with a frequency of 1 Hz. The purpose of the analysis was to determine the glass transition temperature corresponding to the maximum of the tan δ curve. The final temperature of the test was set after the rubbery plateau of the material. The dimension of the samples was set to an average of 0.5 × 3.5 × 12 mm, obtained through curing the formulation into a silicon mold using DYMAX ECE Flood lamp (Dymax Europe GmbH) at a light intensity of 130 mW/cm^2^.

The number of crosslinks per volume *ν_c_* was obtained using Equation (2).
(2)$$v_{c}=\frac{E^{\prime }}{3RT}$$
where *E′* corresponds to the storage modulus in the rubbery plateau (*T_g_* + 50 °C), *T* is the temperature in Kelvin, and *R* is the gas constant.

#### 2.3.4. Tensile Measurements

Mechanical properties of vitrimers were determined through the evaluation of stress–strain curves obtained using a tensile instrument (MTS QTestTM/10 Elite, MTS System Corporation, Eden Prairie, MN, USA) combined with a measurement software (TestWorks^®^ 4, MTS System Corporation). The translation speed was set to 5 mm/min and the dimension of the samples had an average of 2 × 5 × 50 mm^3^. The elastic modulus was determined in the elastic region of the curve (up to 15% deformation). All the results were derived from an average of 5 samples.

#### 2.3.5. Thermogravimetric Analysis (TGA)

Thermal stability was evaluated using thermogravimetric analysis and performed under 50 mL/min nitrogen flow, using a Mettler Toledo instrument equipped with star-e system software. The thermoset was heated from 25 °C to 800 °C, with a heating rate of 10 °C under N_2_ atmosphere.

#### 2.3.6. Dynamic Reversible Network Analysis

Stress relaxation behavior of the UV-cured sample was determined using a Physica MCR 501 rheometer from Anton Paar. The sample dimension had an average dimension of 0.5 mm × 1 mm diameter. The sample was first preloaded with a normal force of 10 N applied for 15 min at the set temperature, then a constant 3% strain was applied and the variation of the relaxation modulus was recorded over time. The chosen temperature corresponded to 170 °C, 180 °C, 190 °C, and 200 °C.

The relaxation modulus *G(t)* was normalized by the initial value *G_t0_*. The vitrimer-characteristic relaxation time was determined as the time needed by the normalized modulus to reach 1/*e*, with an exponential decrease following Equation (3):(3)$$G\left(t\right)=G_{t0\;}e^{\left(-\frac{t}{\tau }\right)}$$

## 3. Results

### 3.1. Photo-Curing Process

The UV-curing process was deeply investigated using two different methods, ATR-FTIR and photo-DSC. In particular, the effect of the photo-initiator concentration (1–4 phr) used in the photocurable formulations on cure rate and final monomer conversion was studied.

The photoinduced ring-opening polymerization reaction was monitored by following the decrease of the epoxy peak area centered around 800 cm^−1^. The area decreased during the light-irradiation time, confirming both the reaction occurring and the formation of crosslinking bonds. Following Equation (1), the percentage of conversion was calculated. The results are shown in Figure 1.

The results show that the epoxy conversion for formulations containing 1 and 2 phr of photo-initiator is similar. In fact, after 60 s of UV irradiation, both formulations reached an epoxy peak conversion of 98%, while in the formulation containing 4 phr of photo-initiator, the final percentage of conversion did not exceed 80%. This could be because at higher concentrations of the photoacid there is already an inner filter effect and a reduction of the incident light which slows down the cure kinetics and final epoxy group conversion.

To confirm the ATR-FTIR analysis, photo-DSC was carried out. The effect of the photo-initiator on ECO conversion was determined following the exothermic peak during irradiation. The graph is reported in Figure 2, while the evaluated integral of the peaks is collected in Table 1.

The photo-DSC analysis confirmed the FTIR data, showing a decrease of the epoxy group conversion for the formulation containing 4 phr of photo-initiator with a lower exothermal peak. On the other hand, it is possible to observe that the formulation with 2 phr of photo-initiator had a higher area, indicating a higher conversion of epoxide groups upon irradiation. Therefore, 2 phr of photo-initiator was chosen as the optimal concentration for curing the resin formulation.

Once settled on the optimal amount of photo-initiator, the influence of the transesterification catalyst on the crosslinking reaction was studied through ATR-FTIR and photo-DSC, with variable amounts of dibutyl phosphate (DP). Based on literature [9,14,50,51], we selected the amount of DP at 10 phr and 15 phr.

Both FTIR and photo-DSC analysis showed a negligible effect of the presence of the DP catalyst in the photocurable formulations up to a content of 15 phr. A very similar epoxy group conversion was observed by FTIR, and an exothermic heat around 256 was recorded for the investigated formulations.

### 3.2. Viscoelastic, Mechanical, and Thermal Properties of Cured ECO-Vitrimers

Viscoelastic properties of the UV-cured samples were tested initially by DMTA analysis. In Figure 3, the DMTA tanδ and E’ curves are reported for the pristine UV-cured ECO samples and for the same sample crosslinked in the presence of 10 phr and 15 phr of the transesterification catalyst DP. Tensile tests were also performed on crosslinked materials and the evaluated elastic modulus are reported in Table 2, together with the number of crosslinks per volume and *T_g_* (taken as the maximum of tanδ).

The thermo-mechanical behavior of the UV-cured sample containing 10 phr DP is slightly increased, both with respect to the pristine UV-cured formulation and the UV-cured formulation containing 15 phr DP, showing that an excess of the transesterification catalyst reduces the crosslinks, decreasing the glass transition temperature, probably because of an early transesterification activation during the curing process.

The thermal stability of UV-cured ECO samples was investigated by TGA analysis reported in Figure 4. Both samples containing the transesterification catalyst showed three degradation steps, while the ECO pristine sample with no catalyst showed only two degradation steps.

Overall, the minimum value of T_d5%_ corresponds to 200 °C, showing excellent thermal resistance of the UV-cured material. The value is significant because the activation of bond exchange reactions and all the reprocessing actions have been investigated, at most, at 200 °C [42].

### 3.3. Dynamic Reversible Network Analysis

The chemical structure of ECO is composed of long and mobile triglyceride chains, which provide both ester and hydroxyl groups that can be exploited for reversible transesterification reactions. At high temperatures, dibutyl phosphate is able to catalyze the transesterification reaction with an exchange of aliphatic chains whose length is determined by ester and hydroxyl groups. The thermo-activated bond exchange reactions and the related change in the viscosity was monitored by stress relaxation experiments, in which the decreasing force, needed to deform the material under a constant strain, is followed over time.

Before testing the material towards vitrimeric behavior, amplitude sweep tests were done to determine the linear viscoelastic region (LVR) and the parameter for the subsequent stress relaxation experiment that had to be conducted in LVR. Amplitude sweep was done at 200 °C, the highest temperature chosen for the stress relaxation experiment (at which no degradation of crosslinked materials occurred according to TGA data), and at 170 °C as the lowest temperature. The samples tested included pristine ECO formulations with no catalyst and crosslinked formulations achieved in the presence of maximum amount of catalyst (15 phr). Recorded amplitude sweep test are reported in Figure 5. It is possible to notice that the linear viscoelastic region extended up to 3% strain, which was applied as deformation for the subsequent stress relaxation tests.

A crucial indicator for the vitrimeric properties is the relaxation time (*τ*), the time needed for the sample’s relaxation modulus *G(t)* to reach 37% (1/*e*) of the initial modulus detected.

The control for the test was chosen as the pristine ECO UV-cured sample since the formulation does not show transesterification exchange reactions due to the absence of the transesterification catalyst. In Figure 6, the stress relaxation for the different UV-cured samples obtained at 200 °C was compared. A certain relaxation can also be observed in the reference UV-cured materials due to the intrinsic viscoelastic behavior of polymeric networks and its high flexibility (low *T_g_*).

On the contrary, the UV-cured samples obtained in the presence of DP showed a marked relaxation behavior. It is possible to notice that not only does the catalyst promote the decreasing of relaxation modulus, but also that a higher amount of DP decreases the relaxation time. The decrease in relaxation time suggests an increase in transesterification rate, and therefore a faster rearranging of the crosslinked structure.

The stress relaxation of UV-cured ECO samples containing 10 phr and 15 phr of DP at different temperatures are reported in Figure 7 and Figure 8, respectively, together with relative Arrhenius plot. Analyzing the figures, it is possible to notice that both UV-cured formulations reached 1/*e* relaxation modulus starting from 170 °C. Relaxation time as a function of temperature can be reported in the Arrhenius graph for both the samples, but while the 15 phr ECO UV-cured sample had a linear fitting the UV-cured sample containing 10 phr, DP showed a change in slope for the Arrhenius plot. Hubbard et al. [52] showed, for the first time, two distinct regions in the Arrhenius plots for vitrimers. The authors reported a kinetic model to explain the change in slope in the Arrhenius graph (that happens in 10 phr ECO UV-cured samples), indicating that, at low temperatures, the role of chain relaxation is faster than the rate of transesterification. Since transesterification must occur prior to chain relaxation, this is the rate-limiting step for stress–relaxation within the network.

## 4. Conclusions

A bio-based epoxy resin derived from castor oil was used to synthetize a dynamic polymer network, with no use of other reactions or solvents. The formulation contained three components: the epoxidized castor oil (ECO), the photo-initiator, and DP as transesterification catalyst. In addition to using a bio-based precursor, the transesterification catalyst is considered as “inherently biodegradable” [53].

ECO showed a high reactivity towards cationic photopolymerization, and the presence of the transesterification catalyst did not significantly influence the photocuring process. The UV-cured networks were fully characterized in term of viscoelastic and hermos-mechanical properties by means of DMTA, tensile test, and TGA analysis. The UV-cured materials containing 10 phr DP showed a slightly higher *T_g_* and mechanical behavior, while by increasing the transesterification catalyst a slight decrease of both *T_g_* and modulus was observed for the dynamic photopolymers.

All investigated samples showed overall thermal stability above 200 °C, and photopolymer networks obtained in the presence of DP facilitated a distinctive relaxation behavior in contrast to a pristine sample crosslinked in the absence of the transesterification catalyst. By increasing the DP content, a decrease of the relaxation time is evident, indicating an increase in transesterification rate, and therefore a faster rearranging of the crosslinked structure.

The stress relaxation of UV-cured ECO samples containing the DP catalysts recorded at different temperatures showed that all the UV-cured formulations reached 1/*e* relaxation modulus starting from 170 °C. With this study, we have demonstrated the possibility of achieving sustainable UV-cured vitrimers by combining bio-based precursors with the use of the environmentally friendly UV-curing technique.

## Figures and Tables

**Figure 1 polymers-15-01024-f001:**
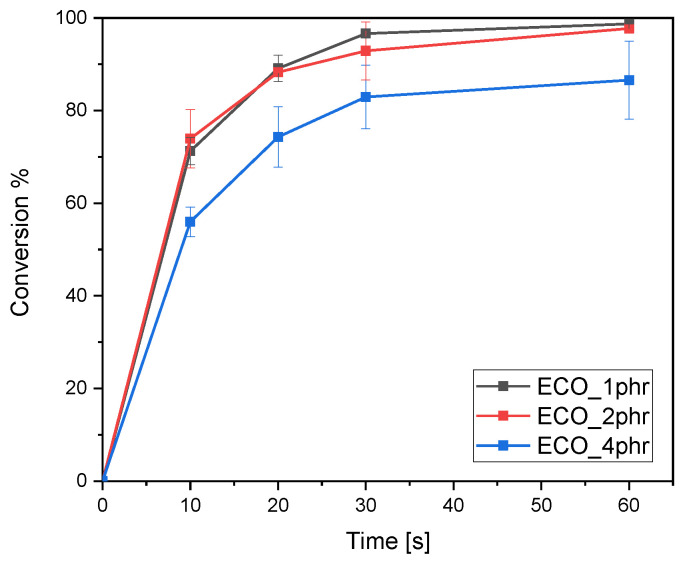
Conversion of epoxy group (from ATR FT-IR) for ECO as a function of time at varying concentration of the photo-initiator. THS photo-initiator was added in 1, 2, and 4 phr. The light intensity was set as 130 W/cm^2^.

**Figure 2 polymers-15-01024-f002:**
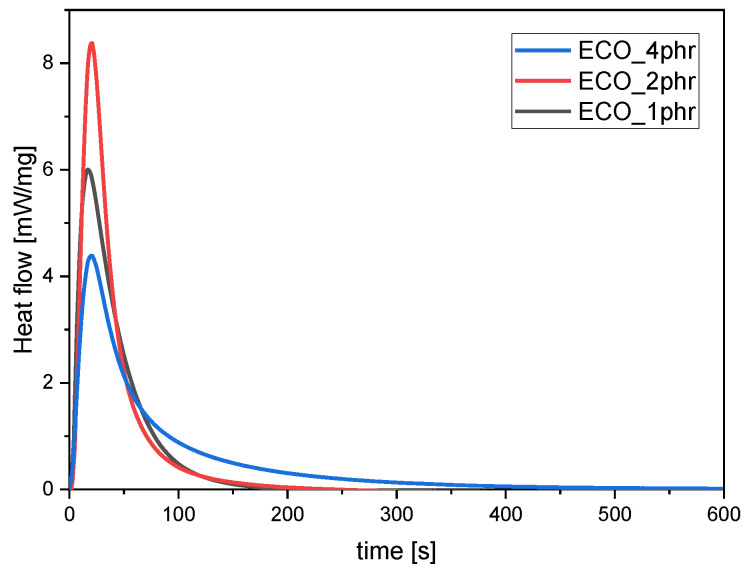
Heat released during UV-light irradiation of ECO as a function of time at varying concentration of photo-initiator. THS photo-initiator was added in 1, 2, and 4 phr. The light intensity was set as 130 W/cm^2^.

**Figure 3 polymers-15-01024-f003:**
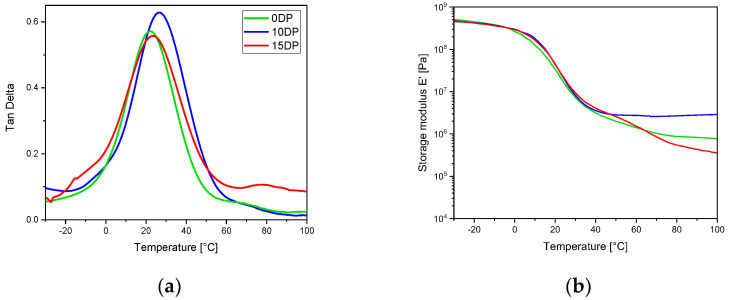
tan δ (**a**) and storage modulus (**b**) for UV-cured samples obtained with different amounts of DP.

**Figure 4 polymers-15-01024-f004:**
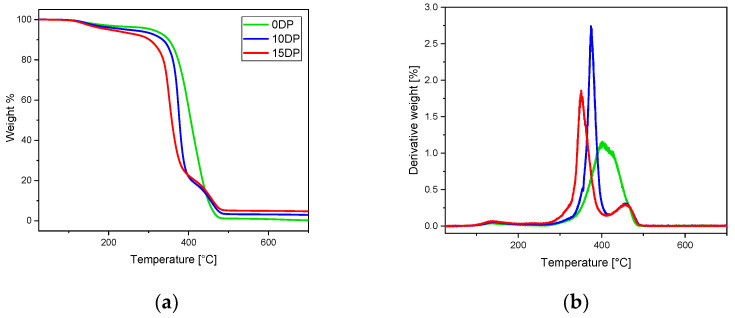
TGA thermal degradation curves of samples with 10 and 15 phr DP and reference sample without DP (**a**) and first derivative of thermal degradation (**b**).

**Figure 5 polymers-15-01024-f005:**
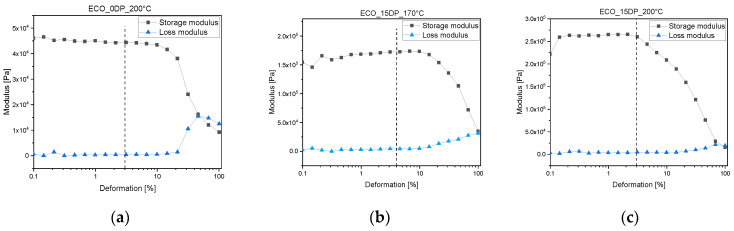
Amplitude sweep tests of pristine ECO UV-cured samples with no catalyst at 200 °C (**a**), ECO UV-cured samples containing 15 phr DP at 170 °C (**b**), and ECO UV-cured samples containing 10 phr at 200 °C (**c**).

**Figure 6 polymers-15-01024-f006:**
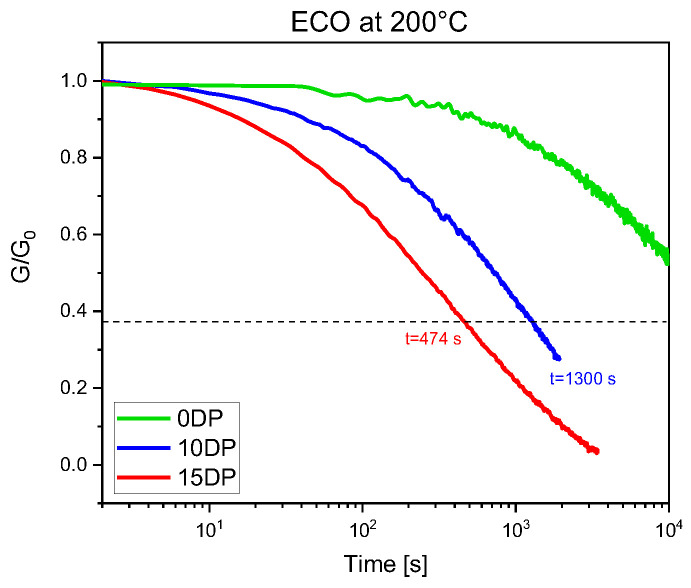
Stress relaxation of ECO with no catalyst, 10 phr DP, and 15 phr DP. The test was done at 200 °C with a constant strain of 3%.

**Figure 7 polymers-15-01024-f007:**
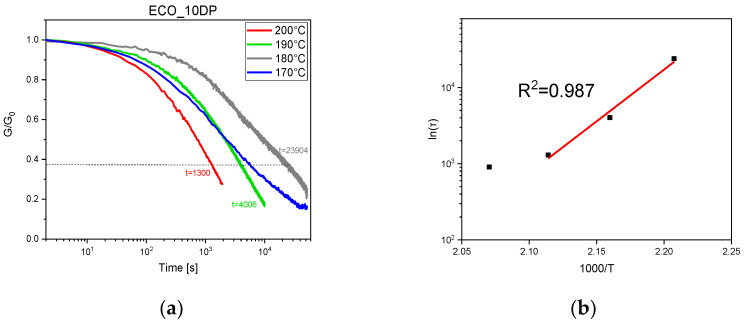
Stress relaxation at 170 °C, 180 °C, 190 °C, and 200 °C of UV-cured ECO samples with 10 phr DP (**a**) and related Arrhenius plots (**b**).

**Figure 8 polymers-15-01024-f008:**
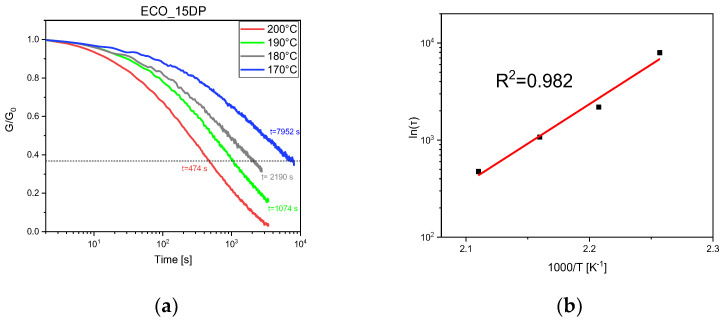
Stress relaxation at 170 °C, 180 °C, 190 °C, and 200 °C of UV-cured ECO samples with 15 phr DP (**a**) and related Arrhenius plot (**b**).

**Table 1 polymers-15-01024-t001:** Heat released during photo-DSC analysis with different amount of photo-initiator.

Sample	Integral [J/g]
ECO_1phr	256.3 ± 15.0
ECO_2phr	296.4 ± 16.7
ECO_4phr	227.1 ± 28.2

**Table 2 polymers-15-01024-t002:** Results obtained by DMTA analysis for samples with different amount of DP. *T_g_* was calculated as the maximum of tanδ. $v_{c}$ calculated by Equation (2).

ECO_2phr	Elastic Modulus [MPa]	$$v_{c}\;[\mathrm{mol}/m^{3}]$$	*T_g_* [°C]
0DP	0.7 ± 0.5	78.3	21 ± 4
10DP	2.7 ± 4.1	307.7	26 ± 4
15DP	0.5 ± 0.1	63.5	23 ± 5

## Data Availability

Not appliable.

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
