# Peer review of "Sustainable Bio-Based UV-Cured Epoxy Vitrimer from Castor Oil"

_polymers, 2023, doi:10.3390/polym15041024_

Round 1

Reviewer 1 Report

Manuscript Number: Polymers-2224747

The manuscript written by Bergoglio et al. titled “Sustainable bio-based UV-cured epoxy vitrimer from castor oil” explored the utility of epoxidized castor oil to produce vitrimeric thermosets that when exposed to high temperature led to change in crosslinked networks. evaluated the effect from the humic acid when incorporated in the EVA/biochar composite.

The article is well written however, I have one major and a few minor questions-

Major

1.   1.   From the data in figure 1, the authors concluded that early vitrification is a result of low epoxy conversion for ECO_4phr. As I know, vitrification occurs due to increase in crosslink density above the point where the network behaves like glass, bust as shown in the figure the conversion for ECO_4phr was lower than ECO_1phr and ECO_2phr. Then how come only ECO_4phr vitrified and not ECO_1phr and ECO_2phr?

Minor

2.    2.  The graphical abstract could be improved for better understanding between changes in the networks before and after transesterification.

3.     3. The introduction could include researches where renewable resources are used to produce thermoset networks like Raut et al (https://doi.org/10.1016/j.cej.2022.139641), Kalita et al. (https://doi.org/10.1016/j.polymer.2021.124191).

4.    4.  What was the EEW of ECO? Please include in sec 2.1.

5.     5. Line 156, change C°/min. to °C/min.

6.     6. Figure 2, ECO_4 phr results in early vitrification, in that case it is strange to observe higher heat release after 100 sec higher than other two formulations. Please add the Y axis title for Figure 2.

7.     7. In figure 3 b, change the Y axis title to storage modulus.

8.     8. Correct  the reference in the table heading for table 2.

Author Response

The responses to reviewer N.1 are rpeorted in the attached file

Reviewer 2 Report

In this manuscript, the author reported the preparation and characterization of bio-based epoxy vitrimer from castor oil. There are some issues need to be addressed, therefore a major revision of this review is recommended.

(1) The sentence of line 15-17 should be rewrote, and the writing of the whole manuscript should be revised.

(2) What is “Vitrimers”? which should be explained in detail in the introduction.

(3) There was almost no references in the result and discussion, how do you know the obtained result is right? And how can you explain the obtained result? Depending guess?

(4) Line 205-207, “probably due an earlier vitrification of the crosslinked polymer.” what is the basis? Is there no any corresponding references?

(5) In Figure 2, the unit, scale and endothermic/exothermic direction for y axis (heat flow) should be added.

(6) Line 248-250, “probably because of an early transesterification activation during curing process.” what is the basis? Is there no any corresponding references?

(7) As shown in Figure 3, the sample containing 10 DP has a higher Tg and crosslinks than the sample containing 0 DP, why does its thermostability is bad (Figure 4)?

(8) The unit and scale should be added in Figure 4b.

(9) Reference 20 and 35 are the same. In addition, almost 30% references were from the authors group.

Author Response

The responses to reviewer N.2 are in the attached file

Round 2

Reviewer 2 Report

The paper was corrected again taking into account all my comments.